# Characterization of Inorganic Scintillator Detectors for Dosimetry in Image-Guided Small Animal Radiotherapy Platforms

**DOI:** 10.3390/cancers15030987

**Published:** 2023-02-03

**Authors:** Ileana Silvestre Patallo, Anna Subiel, Rebecca Carter, Samuel Flynn, Giuseppe Schettino, Andrew Nisbet

**Affiliations:** 1Medical Radiation Physics and Science Groups, National Physical Laboratory (NPL), Guilford TW11 0LW, UK; 2Cancer Institute, University College London, London WC1E 6DD, UK; 3School of Physics and Astronomy, University of Birmingham, Edgbaston Campus, Birmingham B15 2TT, UK; 4Faculty of Engineering and Physical Sciences, University of Surrey, Guildford GU2 7XH, UK; 5Department of Medical Physics & Biomedical Engineering, University College London, Mallet Place Engineering Building, London WC1E 6BT, UK

**Keywords:** radiation detectors, radiation dosimetry, preclinical radiotherapy, X-rays

## Abstract

**Simple Summary:**

Dosimetry for preclinical radiotherapy research requires the standardization of dose validation and quality assurance procedures. Except for dose reference measurements, dosimetric quality control of image-guided small animal irradiation platforms is mostly performed with passive detectors (alanine and gafchromic films). This results in the inconvenient task of lengthy post-processing. In this paper, we carried out a dosimetric characterization of an active detection system based on inorganic scintillators in medium-energy X-rays. We implemented a cross-calibration framework based on international dosimetric protocols to assess the energy dependence of the detector. Additionally, we determined relative output factors for very small fields and compared them to other measurement systems (EBT3 film and CMOS sensor). We demonstrated the suitability of the inorganic scintillation system for the development of phantom-based end-to-end tests for dose verification in newly implemented preclinical radiotherapy irradiation techniques.

**Abstract:**

The purpose of the study was to characterize a detection system based on inorganic scintillators and determine its suitability for dosimetry in preclinical radiation research. Dose rate, linearity, and repeatability of the response (among others) were assessed for medium-energy X-ray beam qualities. The response’s variation with temperature and beam angle incidence was also evaluated. Absorbed dose quality-dependent calibration coefficients, based on a cross-calibration against air kerma secondary standard ionization chambers, were determined. Relative output factors (ROF) for small, collimated fields (≤10 mm × 10 mm) were measured and compared with Gafchromic film and to a CMOS imaging sensor. Independently of the beam quality, the scintillator signal repeatability was adequate and linear with dose. Compared with EBT3 films and CMOS, ROF was within 5% (except for smaller circular fields). We demonstrated that when the detector is cross-calibrated in the user’s beam, it is a useful tool for dosimetry in medium-energy X-rays with small fields delivered by Image-Guided Small Animal Radiotherapy Platforms. It supports the development of procedures for independent “live” dose verification of complex preclinical radiotherapy plans with the possibility to insert the detectors in phantoms.

## 1. Introduction

Preclinical studies are able to accurately model biological and physical aspects of clinical scenarios in radiation oncology. They are critical to the development of personalized and precision medicine and its translational success. The use of preclinical irradiators with image-guidance capabilities, combined with in vivo (tumour) models embedded in mice, offers unique possibilities to cross-examine the radiotherapy response of tumours and normal tissues. Research performed in those scenarios has the potential to translate into improvements in clinical outcomes. Targeted preclinical irradiations are mostly performed using dedicated Image-Guided Small Animal Radiotherapy Platforms (IGSARTP). These devices are capable of delivering realistic radiotherapy plans with beams in the medium-energy X-ray range (0.5–4 mm Cu, in terms of Half Value Layer (HVL)). Due to the size of the targeted regions of interest (ROI), IGSARTP are designed to deliver very small, individual, or combined treatment fields. The complexity of the systems and the need for high accuracy, however, may present additional challenges.

The lack of reproducibility of preclinical data has been reported by several authors. While some of the challenges are related to the specific biology of the preclinical models, there is a consensus on the lack of standardisation and systematic quality assurance for preclinical radiotherapy [1,2]. At the same time, the unavailability of physics support and training for scientists working with biological and preclinical irradiators leads to dosimetric errors and significant variations in the accuracy of the dose delivered between centres and research groups [3,4]. As a result, the comparability and reliability of published results might be compromised [5], and therefore, their translation to novel clinical findings and future benefit to cancer patients might be compromised.

An additional contribution to the above-mentioned difficulties is the fact that current Codes of Practice (CoP) for dosimetry in medium-energy X-rays (IPEMB [6], AAPM TG-61 [7], NCS-10 [8], and IAEA TRS 398 [9]) refer to conditions for reference and relative measurements that are difficult to replicate in the confined spaces of IGSARTP. Determination of the absorbed dose to water at the surface of the phantom or at 2 cm depth (according to the in-air and in-phantom formalisms, respectively) relies on the use of tabulated correction factors that have been established experimentally or by Monte Carlo simulations. The correction factors consider that absorbed dose measurements are performed in irradiation conditions that are typical for clinical kV orthovoltage irradiators, where full backscatter conditions can be achieved. Dosimetry in the medium-energy X-ray range is challenging. One of the reasons relates to the fact that scatter contributions to the dose at points closer to the surface are larger (compared with the MV range) and also significantly dependent on the field size and thickness of the backscatter material. There have been efforts to quantify those effects, but only for fields of 10 mm × 10 mm and larger [10,11].

The complexity of radiotherapy plans delivered by IGSARTP is increasing [12], as are attempts to implement micro-IMRT to deliver clinically analogous treatment techniques [13]. The introduction of spatially fractionated radiotherapy (microbeams) and FLASH, associated with small animal irradiators, is also gaining momentum in preclinical research [14,15]. Validation of dose delivered by preclinical radiotherapy plans is still mostly performed with radiochromic films, alanine, and thermoluminescent dosimeters (TLDs) [3,16,17,18]. These detectors need post-processing and significant expertise for their handling and appropriate analysis of the results. EBT3 Gafchromic film (Ashland Advanced Materials) is the most widespread type of film used for dosimetry in the kV energy range. The latest generation of EBT films shows a better performance in terms of energy dependence than its predecessors; however, the total energy response of EBT3 films irradiated in ^60^Co has been reported to be around 8% larger than the response in a 1.63 mm Cu (HVL) X-ray beam [19]. Working with alanine requires specific infrastructure and equipment (electron paramagnetic resonance (EPR) spectroscopy), which is not widely accessible. Additionally, the absorbed dose to water measured with alanine in the kV X-ray energy range using a ^60^Co calibrated EPR system also requires the application of corrections [20]. While the majority of experimental determinations of the relative response of alanine have been performed in well-characterized X-ray beams, there is still a need for a better understanding of how the relative efficiency is affected when measurements are carried out in small, self-shielded preclinical irradiators [21]. The use of micro ionization chambers, such as the PTW 31022, has also been described for measurements of absorbed dose in preclinical irradiators [22]. Although measurement results can be processed immediately after the irradiation, their use is limited by the larger size of the detector in comparison to the smaller fields used for the delivery of more conformal plans.

Scintillators (organic and inorganic) have been identified as reliable detectors for real- time dosimetry in several radiotherapy applications [23,24,25,26]. One of the challenges for their use as dosimeters is the presence of Cerenkov radiation that is produced during irradiation in the fibre itself, which appears as noise to the detectors’ signal associated with the absorbed dose. Several techniques have been developed to reduce the undesirable effect. Sample optical filtering and chromatic removal are among the most commonly implemented procedures [27]. Cerenkov radiation seems to have less prominence in the response of inorganic scintillators due to their higher light yield [26]. Despite the increased popularity of scintillator detectors for dosimetry among different energy ranges, including those of IGSARTP [28], characterization of the energy dependence in the medium-energy X-ray range has been rarely reported in the literature. The small size of the detectors makes them attractive for measurements in the typically small fields delivered by small animal irradiation platforms.

In this study, the suitability of a commercially available inorganic scintillator, DoseWire (DoseVue, N. V, Diepenbeek, Belgium), for real-time quality assurance (QA) and measurements of dose delivered by preclinical radiotherapy plans was assessed. Similar investigations will be performed for organic scintillators in future studies. Dosimetry characterization of the detector in terms of reproducibility, linearity with dose, evaluation of the response with variation of dose-rate, angle of incidence, as well as beam quality (HVL), was performed at the National Physical Laboratory (NPL), Teddington, UK, employing reference medium-energy X-ray beam qualities (ranging from 0.5 mm Cu to 4 mm Cu). Angular dependence and the feasibility of measuring very small field relative output factors (ROF) as well as the applicability of a quality correction factor for measurements of dose in the end user’s beam (Xstrahl SARRP device) were evaluated.

## 2. Materials and Methods

### 2.1. Inorganic Scintillator-Based Detector

DoseWire (DoseVue, N. V, Diepenbeek, Belgium) consists of a small hemisphere (1 mm diameter) of europium-doped yttrium oxide inorganic scintillating material (Y_2_O_3_:Eu) coupled to an optical fibre. Scintillations are emitted in the 600–650 nm wavelength range. The scintillator material’s effective atomic number (Zeff) is 30.79, and its density is equal to 3.4 g/cm^3^. The detector has a high scintillator light yield and a reduced Cerenkov effect, resulting in a higher signal-to-noise ratio (SNR) [26], compared with those of similar size using organic scintillators.

Figure 1a shows a DoseWire scintillator detector coupled to the optical fibre and subsequently to the robust extension cable, which is connected to the convertor interface. The measurement system characterized in this work has only one channel. The interface is powered by a USB connection to a laptop. 

The signal acquisition workflow is schematized in Figure 1b. The scintillator material undergoes luminescence when irradiated. The light is delivered through the optical fibre to the signal convertor interface box (Figure 1(a.4)), where it is converted into electrical pulses by a photomultiplier tube (PMT) and subsequently to a digital signal. DoseVue (proprietary) software analyses and displays the cumulative number of counts in real time. The information is saved in “plain text” format files.

Three scintillator detectors from two different production batches (all with 1 m of optical fibre) were characterized: DWS1, DWS2, and DWS3.

### 2.2. Experimental Measurements

#### 2.2.1. Detector’s Response with Temperature Variation

The light yield of a scintillator can gradually decrease with increasing temperature. For detectors with potential use in radiotherapy applications, the effect was first reported for BC-400-based plastic scintillators [29]. In preclinical self-contained X-ray cabinets, temperature during the irradiation workflow can rise significantly (from controlled room temperature around 20 °C up to 35 °C, according to our experience). This will have an effect on the temperature inside the phantoms used for dosimetry measurements.

To assess the magnitude of the effect on the DoseWire scintillator, measurements of the stability of the response at different temperatures were performed. The detectors were irradiated (one at a time), submerged in water, inside a T100-R1 refrigerated, heated, circulating bath with a digital control (Stability ±0.1 °C) (Grant Instruments) [30]. Due to practical considerations related to the size of the water bath and the need for the beam to enter vertically through the water reservoir (0° gantry angle), the irradiations were performed at the NPL Theratron 780 (^60^Co) facility (Figure 2). The scintillator was placed at 2 cm depth, centred in a 5 cm × 5 cm field. Five independent measurements of the signal accumulated by the delivery of 2 Gy were recorded at each temperature level, ranging from 19 °C to 35 °C.

A linear response model over the temperature range (Equation (1)) was selected to describe the measured data [31].
(1)$$SS=SS_{0}\left(1+\alpha \left(T-T_{0}\right)\right)$$
where SS is scintillator signal at temperature T, $SS_{0}$ is the signal at the reference temperature $T_{0}$ (20 °C in our case) and α is the temperature coefficient.

#### 2.2.2. Signal Repeatability, Linearity with Dose and Detector Response with the Dose Rate

The scintillators’ signal repeatability, linearity with dose, and response with variation of the dose rate were investigated at four reference medium-energy X-ray qualities (expressed in terms of HVL) using the 300 kV therapy level calibration facility at NPL (see Table 1).

In normal operation conditions, the tube current is set to 10 mA. The facility’s beam line is horizontal and includes a transmission monitor chamber that allows for the verification of the stability of the X-ray source output across all sets of measurements. A bespoke LabVIEW-coded interface was written for accurate control of the shutter’s openings and closings.

Measurements were performed in a 30 cm × 30 cm × 30 cm phantom (slabs with a total of 28 cm of Bart’s WT1 solid water (Phoenix Dosimetry Ltd. n.d.), with the detectors positioned at 2 cm depth. WT1 material has been employed before in preclinical orthovoltage X-ray experiments [33]. In the absence of a customized slab to hold the detector, the scintillators were sandwiched between a pair of 1 cm-thick skinless water equivalent bolus material [34] (Appendix A). In this measurement configuration, the detector’s longitudinal axis was perpendicular to the beam axis. The source for the surface of the phantom (SSD) was set at 75 cm. The field size at the SSD was 7 cm in diameter, and the positioning of the effective point of measurement (EPOM) of the detector at the centre of the radiation field was verified with the Gafchromic RTQA film (Ashland Advance Materials).

At each beam quality, the detectors were irradiated for 30 s. The average of 10 readings was considered for the analysis of the results. The coefficient of variation (CV) in %, expressed as the ratio of the standard deviation of the mean (SD) versus the average, was the metric used to report the short-term repeatability of the response of the scintillators. The coefficient of variation of the readings of the monitor chamber (corrected by pressure and temperature) was also calculated to eliminate variations due to output instability.

The linearity of the response with dose was also assessed for the three detectors. With the tube current at normal operation conditions (10 mA) and for all beam qualities, the detectors were irradiated with doses corresponding to the following irradiation times (in seconds): 5, 20, 30, 60, 120, 240, 500, and 1000. Values of dose corresponding to the irradiation times were previously determined with a traceably calibrated 2611 secondary standard (SS) ionization chamber (S/N 134) and a PTW Unidos electrometer (S/N 0091) performed in the same irradiation conditions as those for measurements with the scintillator (chamber at 2 cm depth in a 30 cm × 30 cm × 30 cm WT1 phantom with an SSD of 75 cm). At each beam quality, the scintillator signal was normalized to the signal at the maximum delivered dose (for 1000 s), and the results were plotted against the irradiation time.

The response of the scintillators as a function of the dose rate was also evaluated. For each quality, the irradiation time was kept constant at 30 s and different dose rates were obtained by changing the X-ray tube current in the range from 2.95 mA to 11.1 mA. The absorbed dose rate at each tube current was established by measurements with the SS 2611 ionization chamber (achieved dose rate range 2.5 to 15 cGy/min). For each beam quality and scintillator, a series of three measurements were acquired. The ratios of the scintillators’ signals versus the monitor chamber’s signals were calculated and normalized against the ratio at the operational tube current of 10 mA. For comparison, the ratios of the SS 2611 IC versus the monitor chamber’s signals were also calculated and normalized to the value of the tube operational current. Results were plotted for all beam qualities, and the coefficient of regression (R) for a linear fit was extracted. 

#### 2.2.3. Angular Response

The investigation of the variation of the scintillator response with the angle of incidence of the beam was performed in Xstrahl’s small animal radiation research platform (SARRP) at the UCL Cancer Institute (CI). The SARRP is one of the two commercially available IGSARTPs. The irradiation platform has a gantry and a robotic stage-based animal positioning system. Both can rotate around the isocentre, which is situated 35 cm from the radiation source. The radiation research platform allows for the delivery of complex radiotherapy plans. Its rotational capabilities allow for a comprehensive assessment of the angular response of the detector.

The first set of measurements was performed in air by placing the scintillator with its longitudinal axis perpendicular to the beam axis. The detector was fixed to the positioning system, and the EPOM was positioned to coincide with the isocentre at the intersection of the lasers and verified with a cone-beam CT image acquisition. The gantry was rotated in increments of 30° until completing a 360° rotation. We will further refer to this measurement configuration as longitudinal angular response.

For the second set of measurements, a bespoke holder was used to place the detector with its longitudinal axis parallel to the radiation beam axis. The detector’s sensitive volume was in the air, and its EPOM was placed at the isocentre. Due to limitations with the positioning holder, the gantry was rotated (clockwise) only from −120° to 120° in 30° steps. This will be further referred to as the polar angular response. Further gantry angles were not investigated as the beam would not be able to reach the detector without interacting first with the holder (Appendix A). 

For both sets of measurements, the 10 mm × 10 mm collimator was used to deliver a 30 s static beam, and three readings of the detector signal were acquired at each gantry angle position. The average of three readings at each gantry angle was normalized to the average of the readings at a 0° gantry angle. Results were plotted in a polar graph with a radial axis representing the variations of the normalized response and the standard deviation of the mean of the three readings.

#### 2.2.4. Energy Dependence and Cross-Calibration in Medium Energy X-ray Beams

Due to the composition of the scintillator material, it is expected for the detector to exhibit a difference in its response to different radiation beam qualities. The energy dependence of the scintillator’s response was studied by irradiating one DoseWire scintillator detector (DWS1) at the previously specified HVLs (Table 1). The cross-calibration was performed against an NPL 2611 secondary standard chamber (SS 2611).

With detectors at 2 cm depth, readings were acquired in solid water (WT1), as described in Section 2.2.2. For each HVL, the following experimental sets were performed:

Measurements with the NPL 2611 secondary standard chamber (SS) and PTW Unidos electrometer, full backscatter (Full_BS) and 60 s for charge collection.

A.Measurements with DWS1, Full_BS, and 30 s for counts collection.B.Measurements with the SS 2611 and PTW Unidos electrometer, 3 cm backscatter (3 cm_BS), and 60 s for charge collectionC.Measurements with DWS1, 3 cm_BS, and 30 s counts collection time.

Measurements with 3 cm backscatter were performed to simulate conditions as recommended by the commissioning protocol set by Xstrahl to determine the beam output of SARRP devices [35].

The absorbed dose to water rate (Gy/s) at a depth of 2 cm in water (or water equivalent phantom), ${\dot{D}}_{w,z=2}$, was calculated from the 2611 chamber measurements, following the recommendations for the in-phantom formalism, as described by the IPEM CoP [6,36].
(2)$${\dot{D}}_{w,z=2}=\dot{M}N_{K}k_{ch}{\left[{\left({\bar{\mu }}_{en}/\rho \right)}_{w/air}\right]}_{z=2,\emptyset }$$
where $\dot{M}$ (charge/s) is the measurement system’s readings corrected for temperature and pressure, $N_{K}$ is the chamber calibration coefficient, ${\left[{\left({\bar{\mu }}_{en}/\rho \right)}_{w/air}\right]}_{z=2,\emptyset }$ is the mass energy absorption coefficient ratio, water to air, averaged over the photon spectrum at 2 cm depth of water and field diameter (subsequently identified by the symbol Ø). Finally, $k_{ch}$ is the correction factor that takes into consideration changes in the response of the ionization chamber that was calibrated in air and is used to perform measurements in water.

A beam quality dependent calibration coefficient $k_{q,Ref\_HVL\left(DWS1\right)}$ (Gy/counts) was determined as the ratio of the ionization chamber measured absorbed dose rate, ${\dot{D}}_{w,\left(SS\right),z=2cm}$ (Gy/s), versus the signal rate from the scintillator ${\dot{Counts}}_{w\left(DWS1\right),z=2cm}$ (counts/s):(3)$$k_{q,Ref\_HVL\left(DWS1\right)}={\dot{D}}_{w,Ref\_HVL\left(SS\right),z=2cm}/{\dot{Counts}}_{w,Ref\_HVL\left(DWS1\right),z=2cm}$$

For every experimental condition: A, B, C, and D, two independent sets of measurements (each with three repeats) were performed for the determination of the $k_{q,Ref\_HVL\left(DWS1\right)}$ Two series of calibration coefficients were determined, $k_{q,Ref\_HVL\left(DWS1\right)\_Full\_BS}$ and $k_{q,Ref\_HVL\left(DWS1\right)\_3cm\_BS}$, for full backscatter and 3 cm backscatter conditions, respectively. 

#### 2.2.5. Cross-Calibration in the User’s Beam

The cross-calibration approach was also performed at UCL’s CI SARRP device (X-ray tube operated at 220 kV and 13 mA), further referred to as the *user’s beam*. With an additional filtration of 0.15 mm Cu and a HVL of 0.667 mm Cu. 

The NPL SS 2611, chamber, and DWS1 were placed at a 2 cm depth in a 30 cm × 30 cm × 5 cm water equivalent phantom (the scintillator was sandwiched between the previously mentioned bolus material). The detectors were irradiated with the 13 cm (side of the equivalent square) open field at an SSD of 33 cm. The position of the detectors’ EPOM was at the centre of the irradiation field.

Following Equation (3), the beam quality-dependent calibration coefficient $k_{q,3cm\_BS,SARRP\_HVL\left(DWS1\right)}^{M@SARRP\_HVL}$ was determined as the ratio of the measured absorbed dose rate ${\dot{D}}_{w,SARRP\_HVL\left(SS\right),z=2cm}$ (Gy/s), versus the signal rate from the scintillator, ${\dot{Counts}}_{w,SARRP\_HVL\left(DWS1\right),z=2cm}$ (counts/s). Its value was compared with the one obtained from the second-degree polynomial fitting of the calibration coefficients determined in non-full backscatter conditions at the four NPL medium-energy X-ray reference beam qualities, $k_{q,3cm\_BS,SARRP\_HVL\left(DWS1\right)}^{Fitted@Ref\_HVL}$.

The process in the *user’s beam* was repeated a different day, but this time the scintillator was cross-calibrated against the traceably calibrated UCL’s SS ionization chamber, PTW 30012 (SN 149). 

For each process, the average of five measurements was taken for the reported values. Each day, two sets of measurements were performed.

#### 2.2.6. SARRP Relative OUTPUT Factor Measurements

To deliver image-guided targeted irradiations, the SARRP at UCL uses individual or combined beams defined by a set of brass collimators with small circular, square, or rectangular fields (up to an area of 10 mm^2^). The measurement of profiles and the relative output of such small fields requires detectors with high spatial resolution. The challenge that these types of measurements pose is highlighted in a review of the commissioning of forty-three IGSARTP [37], which shows a large spread on reported relative dose output factors (ROF) for small circular collimators (e.g., between 0.3–0.8 for 1 mm Ø field). 

The majority of published research refers to the use of film for ROF determination [38,39]. Films have adequate spatial resolution, but the processing and analysis need to be performed off-line. The accuracy of the results depends on having a well-established procedure for handling, scanning, and converting optical density (or pixel value) into values of dose [40].

The three investigated scintillators (DWS1, 2, and 3) were used to measure the SARRP’s ROF. Similar measurements with two other types of detectors were also performed: EBT3 Gafchromic^TM^ films (Ashland, NJ, USA) and a large format 50 mm pixel pitch complementary metal–oxide–semiconductor (CMOS) detector vM1212, “Lassena” [41]. All detectors were positioned at the SARRP’s isocentre, at 2 cm depth in a water equivalent phantom, and 33 cm SSD, with 3 cm backscatter. The ROF obtained with the different types of detectors was compared.

The location of the detectors’ EPOM at the centre of the irradiation field is crucial during small-field measurements. For the scintillators, the position for the maximum signal was verified by continuously irradiating with a 3 mm × 3 mm field (at a 0° gantry angle) while visualizing the number of photon counts at different positions of the robotic stage. At the position of maximum signal (EPOM at the centre of the radiation field), five repeats of 30 s irradiation were acquired, with each of the seven available collimators (Ø 0.5 mm, Ø 1 mm, 3 mm × 3 mm, 3 mm × 9 mm, 5 mm × 5 mm and 10 mm × 10 mm). At the end of each measurement, the total number of photon counts was recorded. The process was repeated for each of the detectors. Two sets of measurements were performed for DSW1.

For measurements with EBT3, film sheets were cut into 8 × 7 cm^2^ pieces for dose calibration and output factor measurements. A nine-point calibration curve (dose range between 0 and 11 Gy) was obtained by irradiating pieces of film in the same conditions as the SS ionization chamber and as described in Section 2.2.5. The dose delivered to each calibration point was calculated based on the previously determined absolute dose rate output: ${\dot{D}}_{w,SARRP\_HVL\left(SS\right),z=2cm}$ (Gy/s). For the ROF measurements, two sets of the same size pieces of film were irradiated for 90 s with the square and rectangular collimators and for 180 s with the smaller circular collimators, respectively. Films were scanned 48 h after the irradiation, using an Epson Expression 10000XL flatbed scanner with 720 DPI resolution. The triple channel method, as implemented in FilmQA Pro^TM^ software (Ashland) was used for the film analysis. Multi-channel dosimetry improves dose accuracy by removing the effects caused by the non-homogeneity of the films and artifacts from the scanner [42]. For each collimator, the values of dose for the red, green, and blue channels in a region of interest (ROI) at the centre of the irradiation field were recorded and averaged.

For measurements with the vM1212 CMOS, a fixed 28 ms integration time was selected to avoid saturation and guarantee the linear response of the detector. The linearity of the relative pixel response was initially calibrated against current measurements with the SS ionization chamber. A corrected image was acquired for each collimator (averaged over a number of frames to reduce noise and with the subtraction of the dark image to eliminate the effect of dark currents) [43]. The detector’s signal (averaged pixel values) within a ROI at the centre of the radiation field was recorded for each collimator.

To be able to compare square, rectangular, and circular fields, the rectangular dimensions of the field sizes were converted into equivalent diameters (Ø in mm) [44].

For each detector, ROF were calculated as:(4)$$ROF_{\emptyset \;x\;\left(mm\right)}=\frac{Measured\;Signal_{\emptyset x\left(mm\right)}}{Measured\;Signal_{\emptyset 11.3\left(mm\right)}}$$
where, $Measured\;Signal_{\emptyset x\left(mm\right)}$ refers to the response of the detector for a given collimator x and $Measured\;Signal_{\emptyset 11.3\left(mm\right)}$, to the response at the reference collimator, in this case the 10 mm × 10 mm (11.3 mm equivalent diameter, Ø).

### 2.3. Uncertainties

Reproducibility, linearity, dose rate dependence, ROF, and directional response Type A uncertainties were determined through the standard deviation of the mean (SDOM) of repeated measurements with a coverage factor of 1 (*k* = 1).

The uncertainty budget of the determination of the scintillator detector HVL-dependent calibration coefficient was derived according to the guide to the expression of uncertainty in measurement from the Joint Committee for Guides in Metrology [45]. The overall uncertainty for the experimental determination of $k_{q,HVL\left(DWS1\right)}$, in the reference and user’s beam qualities is detailed in Table 5, in the Section 3.4.3. The reported uncertainties consider not only variations under the same conditions but also the repositioning of the measurement setup for the three DWS detectors.

## 3. Results

### 3.1. Detectors Response with Temperature Variations

No statistically significant differences in the response to variations in temperature in the range between 19 °C to 35 °C were found for any of the three scintillators. The result is confirmed by the small value of the temperature coefficient α, which on average (for the three scintillators), was 1.14 × 10^−3^ °C^−1^. The largest value for the normalized response $\left(SS/SS_{0}\right)$ was 1.005 for DWS1 and DWS2, at 19 °C, and the lowest was 0.996 at 35 °C for DWS2.

### 3.2. Repeatability, Linearity with Dose and Detector Response with the Dose Rate

Each scintillator has its own response in terms of counts. The ratios of the responses of DWS2 and DWS3 versus DWS1 are 1.1951 ± 0.0016 and 1.0673 ± 0.0012, respectively. The results are averaged over the four beam qualities. The small standard deviation on the ratios demonstrates that independently of a different intrinsic response, each detector has a similar relative energy response. 

On average, over the three detectors and four beam qualities, the signal repeatability is 0.41%, as expressed by the coefficient of variation. The worst repeatability was shown by the DWS2 at 4 mm Cu HVL (CV = 0.88%).

Table 2 presents a summary of the results of the measurements performed to evaluate the repeatability of the scintillators’ response.

All three scintillators showed a linear response with dose in the range up to 2.5 Gy. The results are independent of the beam quality (HVL). Normalized scintillator signal versus dose (expressed in terms of the irradiation time in s) was fitted to a line with a coefficient of regression of R = 0.9999. A graph with dose linearity for all beam qualities is presented in the Appendix A.

The analysis of the measurements to determine the scintillators’ response to dose rate variations (expressed by the tube current) is summarized in Figure 3. On average (considering the responses of the three studied detectors and all beam qualities), the DWS system presents a variation < 2% in the dose rate range achievable by the 300 kV facility at NPL (2.5 to 15 cGy/min). The largest variation (2.9%) was measured for DWS3 at the lower dose rates and for the 4 mm Cu HVL.

### 3.3. Angular Response of the DoseWire

The angular response of the detector is presented in Figure 4. For all gantry angles, the response along the longitudinal axis of the scintillator is within 0.5%. In this first case (Figure 4a), the axis of rotation was parallel to the detector’s axis of symmetry (longitudinal axis), and therefore, a small angular dependence was expected.

A different result was obtained when evaluating the angular dependence while rotating the gantry perpendicular to the axis of symmetry. The maximum difference was found at the extreme angles in both opposite directions with respect to the 0° gantry angle, with a lower signal measured by the scintillator (3.34% and 3.24% difference at 120° and −120°, respectively). The result was expected given the detector’s hemispherical shape (Figure 4b).

### 3.4. Energy Dependence and Scintillator’s Cross-Calibration

#### 3.4.1. Scintillator Detector Calibration Coefficients at NPL Reference Medium Energy X-rays

Table 3 below presents a summary of the measurements acquired at the four NPL medium-energy X-ray qualities, with both the DWS1 and the SS 2611 ionization chambers. The values are the average of three readings.

Figure 5 shows the energy dependence of the DSW1 scintillator, represented by the beam quality calibration coefficient $k_{q,Ref\_HVL\left(DWS1\right)}$, calculated according to Equation (3). For both backscatter conditions: full and 3 cm backscatter (Experiments A–D), values on the graph are the average over the two independent set of measurements.

The largest difference between the calibration coefficients for full and 3 cm backscatter conditions was 4.31% for the hardest beam quality (i.e., 4 mm Cu HVL). The scintillator shows a large energy dependence. For both backscatter conditions, the calibration coefficient is around nine times larger at 4 mm Cu HVL compared with 0.5 mm Cu HVL. For the SS 2611, the same ratio is smaller than 1.02.

#### 3.4.2. Scintillator Cross-Calibration in the User’s Beam

Results of the measurements acquired during the cross-calibration at UCL’s SARRP facility are summarized in Table 4.

The measured HVL-dependent calibration coefficient $k_{q,3cm\_BS,SARRP\_HVL\left(DWS1\right)}^{M@SARRP\_HVL}$, averaged over Set 1 and 2 was 2.643 × 10^−6^ (Gy/counts) and 2.664 × 10^−6^ (Gy/counts) for measurements with NPL SS 2611 and UCL SS PTW 30012, respectively. The difference between the two (0.79%) is within the combined uncertainties of the reproducibility of the setup and daily output variations of the SARRP device, as shown in the uncertainty analysis section. For further comparison and discussion, we considered the average of the two values.

The difference between the averaged measured calibration coefficient, $k_{q,3cm\_BS,SARRP\_HVL\left(DWS1\right)}^{M@SARRP\_HVL}$ (2.654 × 10^−6^, Gy/counts) and that calculated by the polynomial fitting, $k_{q,3cm\_BS,SARRP\_HVL\left(DWS1\right)}^{Fitted@Ref\_HVL}$ (2.999 × 10^−6^ Gy/counts) was −13.02%. That is a larger difference than what was expected. We will further discuss possible causes for the differences.

#### 3.4.3. Uncertainty Budget of the Determination of the Scintillator Detector HVL-Dependent Calibration Coefficient

For the uncertainty analysis, differences in measurement conditions, both at NPL and UCL’s facilities, were considered and summarized in Table 5.

**Table 5 cancers-15-00987-t005:** Measurements recorded during the DWS1 cross-calibration at NPL 300 kV facility.

	NPL 300 kV Facility		UCL SARRP	
Sources of Uncertainty	Type A (%)	Type B (%)	Type A (%)	Type B (%)
Dose at 2 cm depth with SS (Gy/s)		3.2 ^1^		3.63 ^2^
SS charge measurement repeatability (nC)	0.09		0.08	
Temperature (K)	0.02		0.10	
Pressure (kPa)	0.04		0.10	
DWS1 readings repeatability (counts)	1.12 ^3^		0.39 ^4^	
Reproducibility of the setup	1.55 ^5^		1.56 ^5^	
Combined standard uncertainty (k = 1)	3.73		3.97	

^1^ From NPL Report CIRM 41 [46]. ^2^ Adapted from the uncertainty budget for determination of absorbed dose in a SARRP device, considering the use of the 30012 Farmer type ionization chamber (Table 7) [22]. ^3^ NPL 300 kV facility worst case scenario: for 2 mm Cu, 4 mm Cu HVL and 3 cm BS. ^4^ UCL SARRP worst case scenario: Set 2, SS PTW 30012, 3 cm BS. ^5^ Evaluated based on the percentage difference of $k_{q}\;$ values obtained when changing the setup between different Sets. For SARRP it also includes the use of two different SS ionization chambers.

### 3.5. Relative Output Factors

Signals from the three different detectors were analyzed for the different field sizes. For each detector, response at each field size was normalized to that of the 10 mm × 10 mm collimator (11.3 mm equivalent Ø).

Except for the smaller 1 mm and 0.5 mm Ø collimator, ROF was measured with the three DWS agreed within 0.5%. The standard deviation (expressed in terms of the coefficient variation) of four independent determinations of the ROF for the 1 mm Ø was 1.38%, which, considering the size of the scintillator detector (1 mm Ø), was a satisfactory result in terms of the method used to position the detector at the centre of the radiation field. The spread of the results for the 5 mm Ø was significantly larger, 18.59%, as expected, given the larger dimensions of the detector in comparison with the field size.

Tables with the measurements and calculation of the ROF for each detector are provided as Appendix A.

Figure 6 presents the ROF for the DWS as an average of the results for DSW1, 2, and 3. For comparison, the relative output factors measured with EBT3 films and the CMOS Lassena detector are also shown.

ROFs determined with the DWS are lower than those measured with EBT3 and CMOS Lassena. Considering fields with an equivalent diameter (Ø) larger than 5 mm, the differences between ROFs are on average −3.6% and −4.3% when comparing DWS with EBT3 films and CMOS Lassena, respectively. That can be explained by the higher effective atomic number (Z_eff_) of the material of the DSW detectors and differences associated with the energy response at fields with a smaller size than the reference field (10 mm × 10 mm) in the medium-energy X-ray range that are less significant in the interaction with the EBT3 films and the CMOS detector. The same comparison for the 1 mm Ø collimator yields differences larger than 10%. For the smaller 0.5 mm Ø collimator, ROF with the DWSs is two times lower than that measured with the other two detectors. Those results were expected due to the size of the DSW detectors (1 mm Ø) being respectively the same as and larger than the referred field sizes.

## 4. Discussion

No comprehensive characterization of the DoseWire inorganic scintillator system in medium-energy X-rays has been reported. Our study of linearity, dose rate dependence, and repeatability at four reference medium-energy X-ray qualities showed good performance, with adequate short-term repeatability (CV < 0.5%) and excellent linear response with dose in the investigated range (R = 0.9999). The largest response variation with dose rate was 2.9%, for 4 mm Cu HVL, where the signal of the detector was lower. Dose rates in reference conditions of irradiations achieved by SARRP devices (0.0635 Gy/s) are substantially higher than those at NPL 300 kV facility (0.0023 Gy/s). Within the same sampling time, the signal is significantly higher at SARRP, and therefore, smaller variations with dose rate are expected. Each of the available SDW detectors has its own (intrinsic) response (in counts/s) to the same irradiation conditions. The inter-detector’s response variation was expected as the scintillators’ emission characteristics are highly dependent on the manufacturing process, including the crystal growth and doping, its final dimensions, encapsulation, and coupling with the optical fibre. 

The angular response dependence of the detector along its symmetrical axis is negligible. However, along its polar axis, the dependence was larger, at 3.3% at the extreme tested angles (−120° and +120°), decreasing to around 2% for −90° and +90°. That was expected, considering the hemispherical shape of the scintillator. It will be important to take this result into consideration if the device was used to establish protocols for pre-delivery quality assurance or in-vivo verification of complex irradiation plans, including those with non-coplanar beam entries. 

A significant absorbed-dose energy dependence of the inorganic scintillator was expected. This is due to the photoelectric effect, which is the dominant interaction in the medium-energy range, with cross-section values proportional to Z4/E3. The higher Zeff of the Y_2_O_3_/Eu scintillator compared with that of water leads to a significant decrease in the signal measured by the scintillator with an increasing beam energy, as shown by our cross-calibration procedure. The calibration coefficient (expressed in terms of Gy/signal) increases significantly with increasing beam quality (HVL) by dividing the determined absorbed dose to water by the measured signal. With 3 cm backscatter material, the response of the scintillator detector was within 1.4% of that of full backscatter for all reference beam qualities, but for 4 mm Cu HVL, the difference reached 4.1%. The two commercially available IGSARTPs deliver beams with qualities below 1 mm Cu. Our experimental results give an indication that, with a proper cross-calibration process, 3 cm of water equivalent material would provide sufficient backscatter for measurements with the scintillator without significantly increasing the uncertainties of absorbed dose measurements. The significant difference between the fitted and calculated energy dependence calibration coefficient $k_{q}$ could be explained on the basis of the differences in the intrinsic response of the detector to different levels of dose, such as the ones existing between the SARRP and NPL facilities. Although the experimental assessment presents limitations in disassociating the absorbed dose energy dependence from other factors affecting the detector’s response, the lack of appropriate knowledge of the detector’s assembly and components limits the possibility of Monte Carlo modelling.

With all the above, our recommendation will be to cross-calibrate each scintillator detector individually and directly in the user’s beam.

As previously mentioned, published values of ROF for SARRP devices varied significantly and were mostly performed with EBT3 films. There is only one study that reports ROF for the brass collimators measured from images acquired with the SARRP on-board EPID camera-based system [47]. The uncertainty of small-field dosimetry in megavoltage radiotherapy has improved through the compilation of data published by different research groups. One of the most important recommendations is for the experimental determination of ROF to be performed with more than one detector [48]. In order to improve dosimetry verification of IGSARTPs, and considering that the routinely used field sizes in these devices are smaller than those in clinical radiotherapy, the same approach should be considered. The results presented here, to the best of our knowledge, are the first measurements of ROF in medium-energy beams of preclinical irradiators with the use of an inorganic scintillator and the first to compare the results between three different detectors. At this stage, in the case of the scintillators, the ROF was calculated based only on the ratio of the measured detector signals and not on the strict definition that considers the ratio of the absorbed doses. Further experimental work, including analysis of EBT3 films scanned with better resolution and additional measurement sets will help establish the possible need for field size specific correction factors. Those would be able to consider the difference in response of the inorganic scintillator to variations in the contribution of the lower energy components of the X-ray spectra for different fields sizes. Monte Carlo simulations of the response of the detector for that purpose were not possible as the exact composition of the effective volume of the detector was not provided by the manufacturer.

## 5. Conclusions

The dosimetry characterization of the DoseWire scintillator system demonstrates the feasibility of its use for quality assurance purposes and the relative measurements of parameters like the relative output factors of small fields delivered by IGSARTP. The inorganic scintillators have the advantage over films and alanine in that they provide real time measurement results and have smaller dimensions compared with ionization chambers. The methodology described for its cross-calibration contributes to the development of end-to-end tests aiming to achieve independent dosimetry verification of radiotherapy plans delivered by image-guided small animal irradiation platforms. The development of in-vivo dose verification procedures for treatments with larger fields, for example those involving mouse total body irradiation, could expand the potential applications for the use of the inorganic scintillator system not only for IGSARTP but also for conventional preclinical cabinet dosimetry.

## Figures and Tables

**Figure 1 cancers-15-00987-f001:**
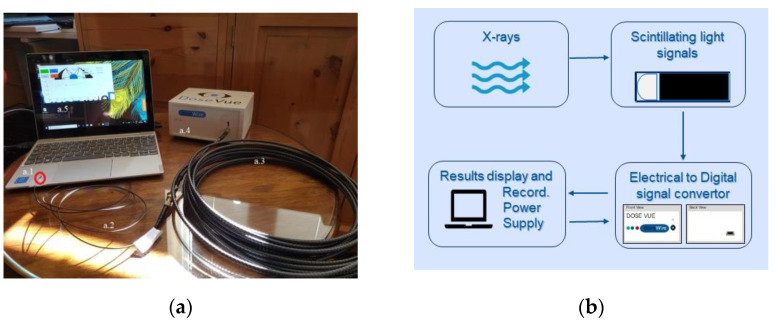
(**a**) DoseVue system: (a.1) DoseWire detector, (a.2) optical fibre, (a.3) extension cable, (a.4) signal convertor interface, (a.5) DoseVue software. (**b**) Signal acquisition workflow.

**Figure 2 cancers-15-00987-f002:**
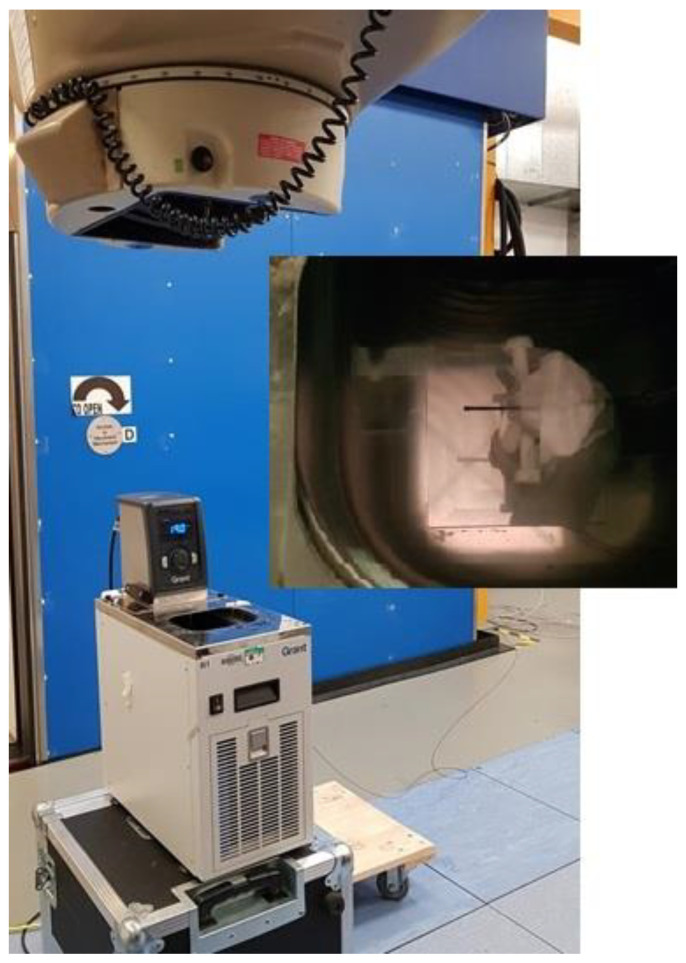
Theratron 780 facility and R1 Grant water bath. A zoomed photograph of the scintillator positioned inside the water bath at the centre of the radiation field.

**Figure 3 cancers-15-00987-f003:**
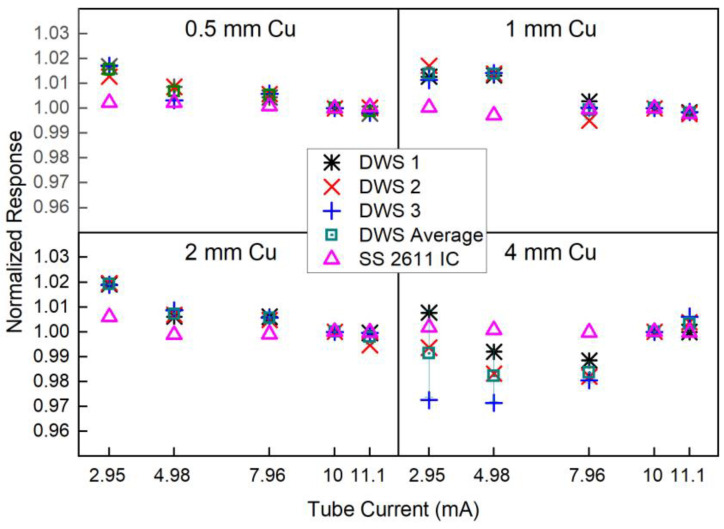
Normalized scintillators’ response to dose rate variation (expressed by changes in the tube current) for different HVLs (0.5, 1, 2 and 4 mm Cu). Errors bars represent the standard deviation of the mean (no visible if smaller than the data point marker).

**Figure 4 cancers-15-00987-f004:**
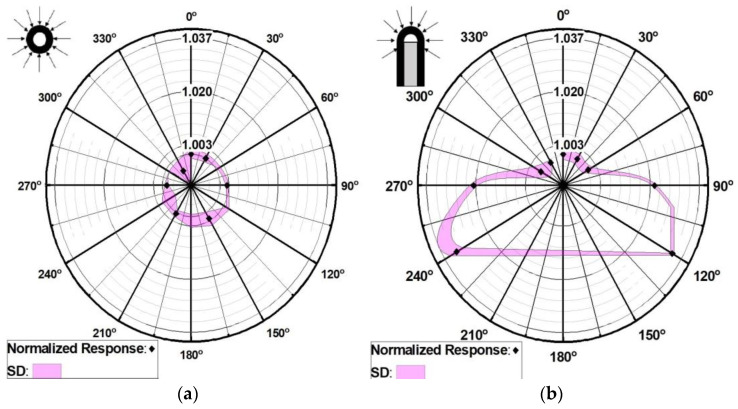
Angular response of DWS 1 scintillator detector. (**a**) longitudinal response, (**b**) polar response. Response is normalized against the signal measured at SARRP 0° gantry angle. SD correspond to the standard deviation of three readings at each gantry angle.

**Figure 5 cancers-15-00987-f005:**
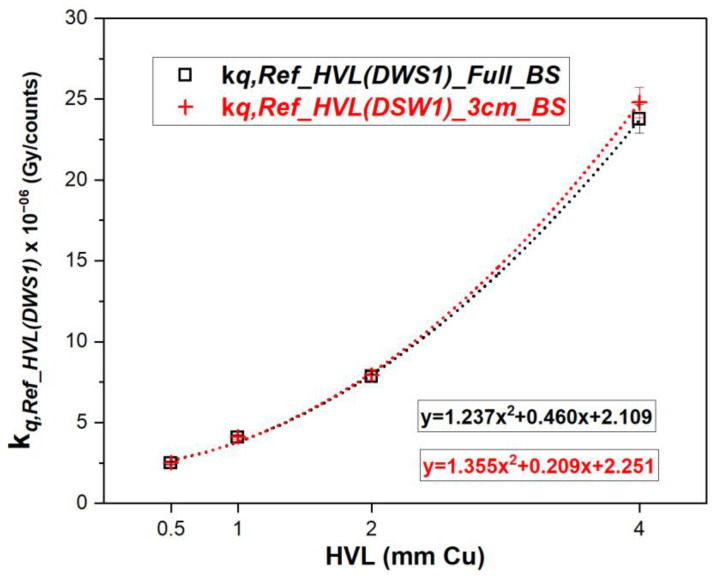
Beam quality dependent calibration coefficient, kq, determined in conditions of full and 3 cm backscatter, respectively. Error bars stand for the combined standard uncertainty (k = 1). When not visible, the value is smaller than the data point marker.

**Figure 6 cancers-15-00987-f006:**
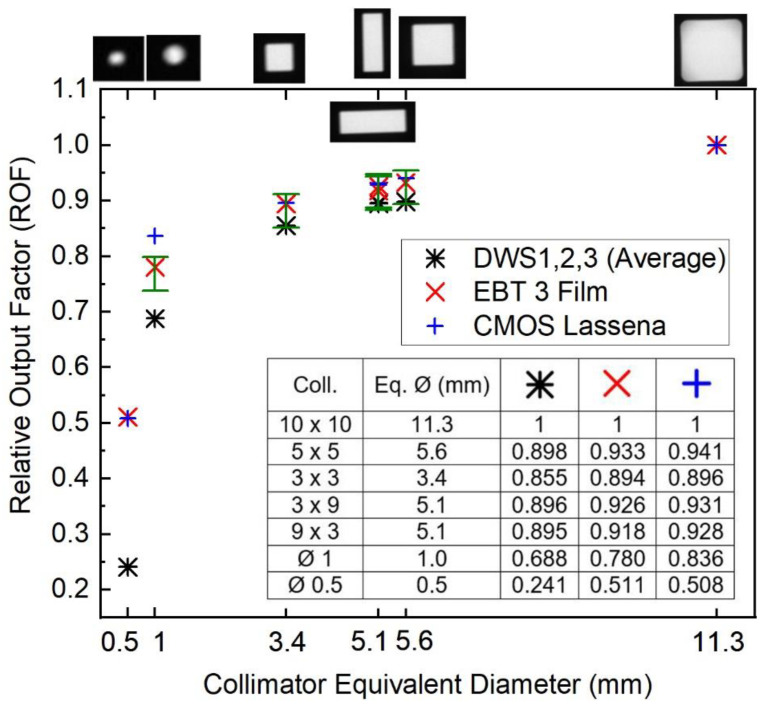
ROF for SARRP brass collimators. Detectors’ EPOM positioned at 2 cm depth, at the isocentre, 3 cm backscatter, and 33 cm SSD. Error bars represents a fixed standard deviation of ±3% for ROF, on the average over the three different detectors. Representative images of the different field shapes and sizes (shown on the upper horizontal axis) were acquired with the CMOS Lassena detector and are not at scale.

**Table 1 cancers-15-00987-t001:** NPL therapy level (medium-energy X-rays) reference qualities.

Nominal Generating Potential (kV)	Additional Filtration (mm Sn + mm Cu + mm Al) ^1^	HVL (mm Cu)	
		First	Second ^2^
135	0 + 0.27 + 1.2	0.50	0.85
180	0 + 0.54 + 1.0	1.00	1.69
220	0 + 1.40 + 0.9	2.00	2.87
280	1.5 + 0.26 + 1.0	4.00	4.39

^1^ The X-ray tube has an inherent filtration of 0.3 mm beryllium plus 4.8 mm of PMMA. ^2^ Second HVL determined by SpekCalc calculations [32].

**Table 2 cancers-15-00987-t002:** Repeatability measurements for the scintillator detectors.

		0.5 mm Cu	1 mm Cu	2 mm Cu	4 mm Cu
Scintillator (counts) ^1^	DWS1	21,057.36	19,765.84	10,028.23	3205.56
	DWS2	25,115.18	23,626.34	12,005.56	3831.85
	DWS3	22,514.24	21,097.56	10,699.68	3416.74
Standard Deviation	DWS1	84.74	56.65	33.40	24.40
	DWS2	83.83	58.13	34.78	33.76
	DWS3	82.25	57.21	34.62	16.43
Coefficient of variation (%)	DWS1	0.40	0.29	0.33	0.76
	DWS2	0.33	0.25	0.29	0.88
	DWS3	0.37	0.27	0.32	0.48
Monitor IC (µC) ^1^		0.047055	0.067651	0.066751	0.065559
Standard Deviation		0.000014	0.000030	0.000029	0.000025
Coefficient of variation (%)		0.03	0.04	0.04	0.04

^1^ Average of 10 readings, each acquired for 30 s.

**Table 3 cancers-15-00987-t003:** Measurements recorded during the DWS1 cross-calibration at NPL 300 kV facility.

	0.5 mm Cu	1 mm Cu	2 mm Cu	4 mm Cu
	Set 1: Full Backscatter (A and B)			
SS 2611 average dose rate (Gy/s)	1.745 ×10^−3^	2.564 × 10^−3^	2.464 × 10^−3^	2.335 × 10^−3^
DWS1 average signal (counts/s)	700.98	626.00	311.90	97.37
	Set 1: Full Backscatter (A and B)			
SS 2611 average dose rate (Gy/s)	1.748 × 10^−3^	2.560 × 10^−3^	2.460 × 10^−3^	2.332 × 10^−3^
DWS1 average signal (counts/s)	702.10	628.70	316.23	99.02
	Set 1: Full Backscatter (A and B)			
SS 2611 average dose rate (Gy/s)	1.690 × 10^−3^	2.499 × 10^−3^	2.460 × 10^−3^	2.302 × 10^−3^
DWS1 average signal (counts/s)	668.62	618.51	304.97	92.41
	Set 1: Full Backscatter (A and B)			
SS 2611 average dose rate (Gy/s)	1.688 × 10^−3^	2.495 × 10^−3^	2.421 × 10^−3^	2.294 × 10^−3^
DWS1 average signal (counts/s)	669.45	622.39	306.11	93.03

**Table 4 cancers-15-00987-t004:** Measurements recorded during the DWS1 cross-calibration at UCL’s SARRP facility.

		0.667 mm Cu
Day 1	SS 2611 average dose rate (Gy/s) Set 1	6.254 × 10^−2^
	DWS1 average signal (counts/s) Set 1	23,620.03
	SS 2611 average dose rate (Gy/s) Set 2	6.314 × 10^−2^
	DWS1 average signal (counts/s) Set 2	23,927.97
Day 2	SS PTW 30012 average dose rate (Gy/s) Set 1	6.242 × 10^−2^
	DWS1 average signal (counts/s) Set 1	23,486.45
	SS PTW 30012 average dose rate (Gy/s) Set 2	6.333 × 10^−2^
	DWS1 average signal (counts/s) Set 2	23,710.09

## Data Availability

Data supporting reported results will be find in can be found within Zenodo-Research. Shared (data will be linked to DOI assigned to the paper).

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
