# Peer review of "Characterization of Inorganic Scintillator Detectors for Dosimetry in Image-Guided Small Animal Radiotherapy Platforms"

_cancers, 2023, doi:10.3390/cancers15030987_

Round 1

Reviewer 1 Report

Your paper describing the performance characteristics of a scintillator based dosimetry system is very thorough, comprehensive and well written.  The thoroughness of the work does lead to there being a large number of pages between the description of the various tests and investigations and the results which requires the reader to keep moving back and forth between sections, but I guess not much can be done about that!

There are two minor clarifications, however, that I think would add to the value of the paper.

Lines 137-139: Subsequently the light signal is converted to a digital signal, which is collected and analysed in real time by the DoseVue (proprietary) software which displays (and saves in “plain text” format) the cumulative number of counts. 

Are you able to describe the process of converting the light signal to a digital signal in more detail? Are avalanche photo diodes or a pmt used?  Or is it a current measureing system that is subsequently converted to a digital signal?

Lines 520-522: A significant absorbed dose energy dependence of the inorganic scintillator was expected. Due to the high Zeff, the photoelectric effect in the Y2O3/Eu detector material is dominant across the medium energy x-ray range and causes the significant changes in the detector’s response as a function of the beam energy. 

More explanation here would help the reader understand this relationship. As the X-ray energy increases (increasing HVL) the calibration coefficient kq increases, presumably due to the scintillator becoming more efficient as a result of photoelectric cross section increasing with decreasing photon energy. 

Author Response

Thank you to the reviewer for the time dedicated to review the manuscript and for the positive feedback provided. I have addressed the comments as below:

Lines 137-139: Subsequently the light signal is converted to a digital signal, which is collected and analysed in real time by the DoseVue (proprietary) software which displays (and saves in “plain text” format) the cumulative number of counts. 

Are you able to describe the process of converting the light signal to a digital signal in more detail? Are avalanche photo diodes or a pmt used?  Or is it a current measuring system that is subsequently converted to a digital signal?

In relation to the clarifications required, please find below the changes to the manuscript:

Lines 135-142 (now 144-149), the paragraph now reads:

The scintillator material undergoes luminescence when irradiated. The light is delivered through the optical fibre to the signal convertor interface box (Figure 1. (a.4)), where it is converted into electrical pulses by a photomultiplier tube (PMT) and subsequently to a digital signal. DoseVue (proprietary) software analyses and displays in real time, the cumulative number of counts. The information is saved in “plain text” format files.

For the second comment

Lines 520-522: A significant absorbed dose energy dependence of the inorganic scintillator was expected. Due to the high Zeff, the photoelectric effect in the Y2O3/Eu detector material is dominant across the medium energy x-ray range and causes the significant changes in the detector’s response as a function of the beam energy. 

More explanation here would help the reader understand this relationship. As the X-ray energy increases (increasing HVL) the calibration coefficient kq increases, presumably due to the scintillator becoming more efficient as a result of photoelectric cross section increasing with decreasing photon energy. 

I changed the paragraph, that now reads:

Lines 533-540 A significant absorbed dose energy dependence of the inorganic scintillator was expected. This is due to photoelectric effect, which is the dominant interaction in the medium energy range, with cross section values proportional to Z4/E3. The higher Zeff of the Y2O3/Eu scintillator, compared to that of water leads to a significant decrease in the signal measured by the scintillator with an increasing beam energy, as shown by our cross-calibration procedure. The ratio of the determined absorbed dose to water to the measured signal, leads to a calibration coefficient (expressed in terms of Gy/signal), that significantly increases with the increase of the beam quality (HVL).

Reviewer 2 Report

The usage of an inorganic scintillator for radiotherapy measurements is presented in this paper. The parameters were compared between 3 detectors and with an ion chamber. The dosimetry accuracy is discussed. The study showed a well-established potential of using scintillators in small radiation field radiotherapy. The paper is well organized and all tables and graphs support the conclusions.    

Minor corrections:

1. Line 137: "Subsequently the light signal is converted to a digital signal..." - please elaborate on what is the convertor (PMT, or photo-diode).

2. In Figure 1 the label (a) appears in both images.

Author Response

Reviewer 2

Thank you to the reviewer for the time dedicated to review the manuscript and for the positive feedback provided.

In relation to the clarifications required, please find below the changes to the manuscript:

  1. Line 137: "Subsequently the light signal is converted to a digital signal..." - please elaborate on what is the convertor (PMT, or photo-diode).

I changed the paragraph that now reads:

Lines 135-142 (now 144-149), the paragraph now reads:

The scintillator material undergoes luminescence when irradiated. The light is delivered through the optical fibre to the signal convertor interface box (Figure 1. (a.4)), where it is converted into electrical pulses by a photomultiplier tube (PMT) and subsequently to a digital signal. DoseVue (proprietary) software analyses and displays in real time, the cumulative number of counts. The information is saved in “plain text” format files.

  1. In Figure 1 the label (a) appears in both images.

I changed the label to (b)

Reviewer 3 Report

This paper is interesting and should be published. I have a few major comments that should be addressed prior to publication.

General comments

The detection system is based upon inorganic scintillators. This is okay for this paper; however, why not also look at organic scintillators, which are also available through a manufacturer. Organic scintillators are mentioned but not explored. I guess this would be an intent for another publication.

Table 1 and in the text, HVL is mentioned. However, to adequately describe the beam in question, the homogeneity coefficient (or second HVL) should be measured or given. The homogeneity coefficient is an aid to describe different x-ray spectra.

Since a scintillator is one of the major points in this paper, it is very important that a discussion and measurement of the Cerenkov signal be done. This is a major aspect missing in the manuscript and must be included before publication.

Minor comments:

1.     Line 65 the tense for compromise is wrong. It should be compromised’

2.     Line 86 Beside radiochromic film and alanine, TLDs are in use for interinstitutional measurements. _ See your reference 3 and also the publication: IJROBP Vol 111, No. 5 pp e75-e81 (2021) – Accurate Dosimetry for Radiobiology.

3.     Line 90-92 Variations have (not has). I would prefer a reconstruction of this sentence for better clarity.

4.     Figure 1 There are two (a)s – no (b) in figure

5.     Line 211 Scintillator, a series … Add a

6.     Line 268 Definition of circle with slash, used for diameter. See line 326. This is important to define it the first time used. Instead of the symbol use words here to define how you will use it and explain what you want to convey by its use.

Author Response

Thank you to the reviewer for the time dedicated to review the manuscript and for the constructive feedback provided.

All the comments and suggestions are addressed and changed in the manuscript, as presented in the attached document with all the responses.
